# A Case of Hepatic Immunoglobulin G4-Related Disease Presenting as an Inflammatory Pseudotumor and Sclerosing Cholangitis

**DOI:** 10.3390/diagnostics12061497

**Published:** 2022-06-19

**Authors:** Se Young Jang, Young Seok Han, Sang Yub Lee, Ja Ryung Han, Young Oh Kweon, Won Young Tak, Soo Young Park, Yu Rim Lee, Hun Kyu Ryeom, Jung Guen Cha, Jihoon Hong, Yoo Na Kang

**Affiliations:** 1Department of Internal Medicine, School of Medicine, Kyungpook National University, Kyungpook National University Hospital, 130 Dongdeok-ro, Jung-gu, Daegu 41944, Korea; vocjsy@knu.ac.kr (S.Y.J.); yokweon@knu.ac.kr (Y.O.K.); wytak@knu.ac.kr (W.Y.T.); psyoung@knu.ac.kr (S.Y.P.); deblue00@naver.com (Y.R.L.); 2Department of Surgery, School of Medicine, Kyungpook National University, Kyungpook National University Hospital, 130 Dongdeok-ro, Jung-gu, Daegu 41944, Korea; gshys@knu.ac.kr (Y.S.H.); jh40356@gmail.com (J.R.H.); 3Department of Radiology, School of Medicine, Kyungpook National University, Kyungpook National University Hospital, 130 Dongdeok-ro, Jung-gu, Daegu 41944, Korea; sangyub@knu.ac.kr (S.Y.L.); hkryeom@knu.ac.kr (H.K.R.); ircha7527@knu.ac.kr (J.G.C.); blushain@knu.ac.kr (J.H.); 4Department of Forensic Medicine, School of Medicine, Kyungpook National University, 680 Gukchaebosang-ro, Jung-gu, Daegu 41944, Korea

**Keywords:** immunoglobulin G4-related disease, cholangitis, sclerosing, immunohistochemistry, biopsy, liver, surgery

## Abstract

An inflammatory pseudotumor is a benign disease characterized by tumor-like lesions consisting of inflammatory cells including plasma cells and fibrous tissue. Recently, some inflammatory pseudotumor cases proved to be a form of Immunoglobulin G4-related disease (IgG4-RD). This novel clinical entity, recognized as a fibroinflammatory condition, is characterized by lymphoplasmacytic infiltration with a predominance of IgG4-positive plasma cells, storiform fibrosis, and often elevated serum IgG4 concentrations. We report a case of IgG4-RD in the form of an inflammatory pseudotumor in the liver with combined sclerosing cholangitis. We recommend that for diagnosing IgG4-RD accurately, it is important to obtain adequate tissue samples and follow-up the lesion in clinical practice.

An inflammatory pseudotumor is characterized by tumor-like lesions consisting of inflammatory cells and fibrous tissue. An inflammatory pseudotumor was first described in the lung [1], but is now known to be found in nearly every organ in the body. The etiology and pathophysiology of inflammatory pseudotumors are not yet clearly understood. However, infection by bacteria such as *Klebsiella pneumoniae* and *E. coli* [2,3] and viruses such as Epstein-Barr virus [4], or the presence of inflammatory conditions including Immunoglobulin G4-related disease (IgG4-RD) can explain the development of this entity. Recently, many cases of inflammatory pseudotumors due to IgG4-RD have been reported. 

IgG4-RD is a novel clinical entity recognized as a fibroinflammatory condition, characterized by lymphoplasmacytic infiltration with a predominance of IgG4-positive plasma cells, storiform fibrosis, and often elevated serum IgG4 concentrations [5]. IgG4-RD can affect multiple organs [6,7], such as the pancreas, retroperitoneum, kidneys, and bile ducts. It presents as a multi-organ disease and can be confused with malignancy, infection, or other immune-mediated conditions [8]. IgG4- RD can lead to organ dysfunction and organ failure due to late recognition and diagnosis of the disease [9].

This paper reports the case of IgG4-RD of the liver and bile ducts presenting as an inflammatory pseudotumor. 

A 62-year-old male was referred to the department of hepatobiliary surgery in our hospital with a newly developed liver mass and right back pain after laparoscopic cholecystectomy. A month before, the patient had visited the emergency room complaining of fever and right upper quadrant (RUQ) pain. His laboratory findings showed elevated white blood cell counts (11.2 (reference: 4.8–10.8) × 10^3^/µL), elevated aspartate transaminase (128 U/L (reference: ≤40)), elevated alanine transaminase (96 U/L (reference: ≤41)), elevated alkaline phosphatase (180 U/L (reference: 40–129)), elevated gamma-glutamyltransferase (132 U/L (reference: 10–71)), and slightly elevated total bilirubin (1.8 mg/dL (reference: <1.2)). He was diagnosed with acute cholangitis and he had stones removed from the common bile duct. Following laparoscopic cholecystectomy, pain in RUQ and back developed. Computed tomography (CT) showed a newly developed 6 cm-sized mass in segments 6 and 7 of the liver as compared with CT scan images before cholecystectomy (Figure 1). He was referred for further evaluation of the hepatic lesion. Laboratory tests showed an elevated erythrocyte sedimentation rate (88 mm/h (reference: 0–10)) and c-reactive protein (6.63 mg/dL (reference: <0.5)) without other laboratory abnormalities. Repeated CT showed the ill-defined mass with upstream bile duct dilatation (Figure 2). The hepatic lesion was biopsied with a 16 gauge-needle under ultrasonography (US) guidance. Final diagnosis was an inflammatory pseudotumor. 

On the 6-month follow-up liver dynamic CT (Figure 3A–C), there was a still mass on liver segments 6 and 7. In addition, aggravated right intrahepatic duct dilatation and new low-density nodules appeared in segment 6. Bile ducts were more dilated compared with the previous CT scan taken at the time of needle biopsy.

We hypothesized that there might be biliary strictures causing upstream bile duct dilatation, recurrent cholangitis, and consequently a liver abscess. Due to claustrophobia, the patient did not undergo magnetic resonance imaging. A US-guided right posterior bile duct access cholangiography was done to explore the lesion and perform therapeutic bile drainage with recanalization. However, this trial failed, showing focal tortuous distension of the bile duct with outflow obstruction (Figure 3D). We decided to resect the tumor that was suspected to be an infected early-stage biloma with the bile duct, causing recurrent infections. Before resection, CT showed that the size was enlarged by 6.4 cm with an increased inner cystic lesion along with stationary bile duct dilation (Figure 4). Finally, the hepatic resection was performed.

We reviewed the pathology of the resected liver with hematoxylin & eosin staining (Figure 5A–C). Diffuse lymphoplasmacytic infiltrates with lymphoid aggregates in a fibrotic background were present. Obliterative phlebitis and storiform fibrosis were observed. The findings were suspicious of IgG4-RD; therefore, immunohistochemistry of IgG4 and IgG was performed for diagnosis (Figure 5D). IgG4-RD involving the liver and bile ducts was diagnosed. The serum IgG4 checked after resection was 749.1 mg/L (reference: 30–2010).

After liver resection, the patient started taking 0.5 mg/kg prednisolone once a day. However, he complained of moon face and was allergic to azathioprine substituting for prednisolone; thus, we discontinued these medications. He underwent periodic laboratory and imaging tests. The 6-month follow-up CT showed no remnant of the mass nor newly developed bile duct dilatation or masses. Informed consent was obtained from the patient regarding the publication of this case report. 

To our knowledge, this is the first report of a case of abrupt onset and progression of an inflammatory pseudotumor caused by IgG4-RD involving the liver and bile ducts. There have been some reported cases of IgG4-related hepatic inflammatory pseudotumors or IgG4-related sclerosing cholangitis (IgG4-SC) mimicking cholangiocarinoma [10,11,12]. However, this case is unique because IgG4-RD developed not only in the liver parenchyma but also in the bile ducts, presenting as stenosis and stricture, which can be diagnosed as IgG4-SC. According to the literature [13,14], sclerosing cholangitis is a condition of progressive stenosis and destruction of the bile ducts because of diffuse inflammation and fibrosis and includes IgG4-SC. 

In addition, in our case, the patient’s newly developed mass was biopsied. Prior to biopsy, it looked like a mass-forming intrahepatic cholangiocarcinoma with bile duct dilatation. This mass lesion had abruptly developed in one month. However, pathology of 3 pieces measuring 1.5 cm in length each, showed benign lesions with non-specific inflammation around the bile duct. 

In this case, serum IgG4 was not elevated after resection. It could have been in the middle of falling or the initial serum IgG4 level was in a normal range. It has been recently recognized that serum IgG4 levels are not elevated in a substantial percentage of patients diagnosed as IgG4-RD and serum IgG4 elevation is no longer considered essential to the diagnosis of IgG4-RD [15]. In an IgG4-RD of inflammatory pseudotumor, as the imaging and laboratory findings are not pathognomonic, needle biopsy is widely performed to get the clue for diagnosis. However, needle biopsy has shown to be inferior in obtaining a histological diagnosis of IgG4-RD as compared to surgical biopsy, especially in detecting the IgG4/HPF count [16]. In this case of IgG4-RD of the liver, needle biopsy also showed a limitation in diagnosing IgG4-RD.

Based on the recent introduction of a novel clinical entity of IgG4-RD, it draws more attention and makes the diagnostic rate higher. Even if histological findings are necessary to diagnose IgG4-RD, it is sometimes difficult to obtain adequate tissue samples. In our case, although needle biopsy was performed by an expert to obtain 3 tissue samples, it showed no evidence of IgG4-RD. The reason might be that the lesion was at a very early stage of development, showing heterogeneity. Thus, if a mass is suspicious of a benign inflammatory and fibrous lesion, immunohistochemical staining for IgG and IgG4 should be done to exclude IgG4-RD.

We followed the case of IgG4-RD manifesting as a hepatic inflammatory pseudotumor diagnosed and resolved by hepatic resection with histopathological examination. In conclusion, it is important to suspect IgG4-RD for mass lesion early and to diagnose IgG4-RD accurately by obtaining adequate tissue samples and following-up the lesion.

## Figures and Tables

**Figure 1 diagnostics-12-01497-f001:**
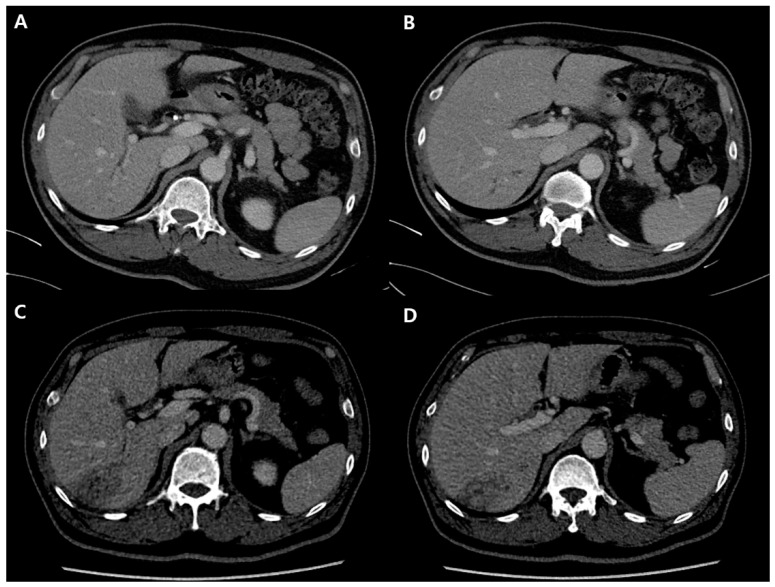
Before (**A**,**B**) and 1 month (**C**,**D**) after laparoscopic cholecystectomy. Multiple, focal bile duct dilation is shown (**A**,**B**). The newly developed low density mass overlays in segment 6 and 7 (**C**,**D**).

**Figure 2 diagnostics-12-01497-f002:**
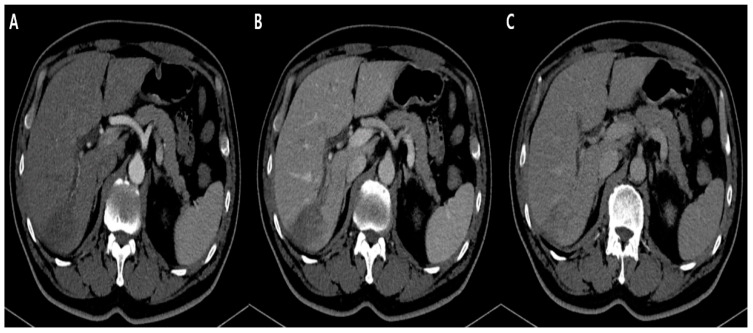
Arterial phase (**A**), portal phase (**B**), and delayed phase (**C**) of axial CT scan shows a poorly enhancing low density mass with right upstream bile duct dilatation.

**Figure 3 diagnostics-12-01497-f003:**
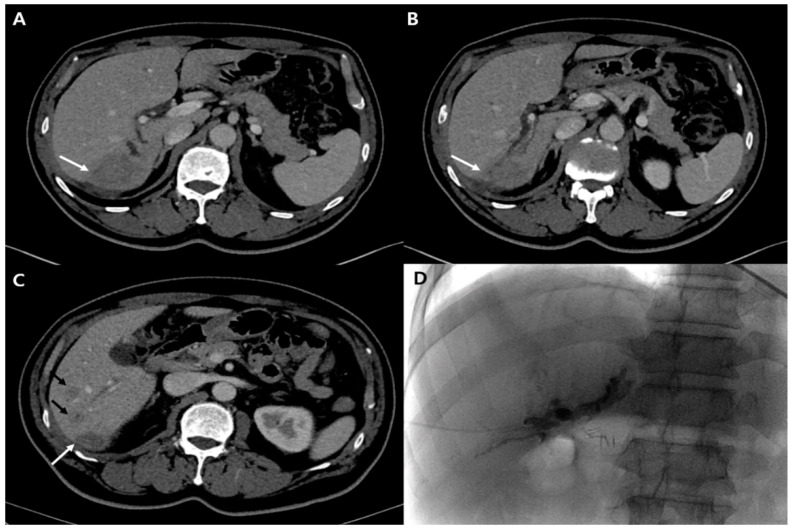
Serial portal phase CT scan (**A**–**C**) shows aggravated right intrahepatic duct dilatation and newly appeared low density nodules in the liver segment 6 (**C**, black arrows). Main low density mass (**A**–**C**, white arrows) size was not changed. Percutaneous biliary drainage was performed via right posterior bile duct. Catheter passage to the downstream was not possible for the tight biliary stricture at biliary confluence (**D**).

**Figure 4 diagnostics-12-01497-f004:**
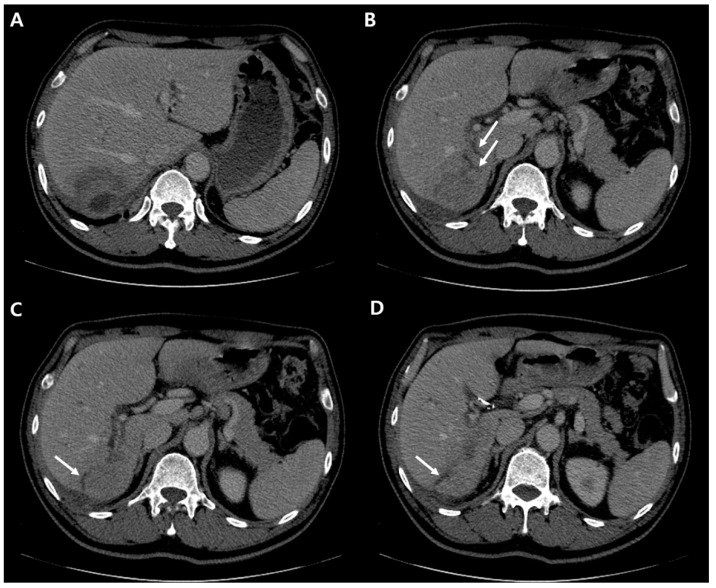
The mass on segment 7 has increased in size and the inner cystic area has newly developed (**A**). Before the hepatic resection, CT scan of portal phase demonstrates stationary state of biliary dilatation (**B**–**D**, white arrows).

**Figure 5 diagnostics-12-01497-f005:**
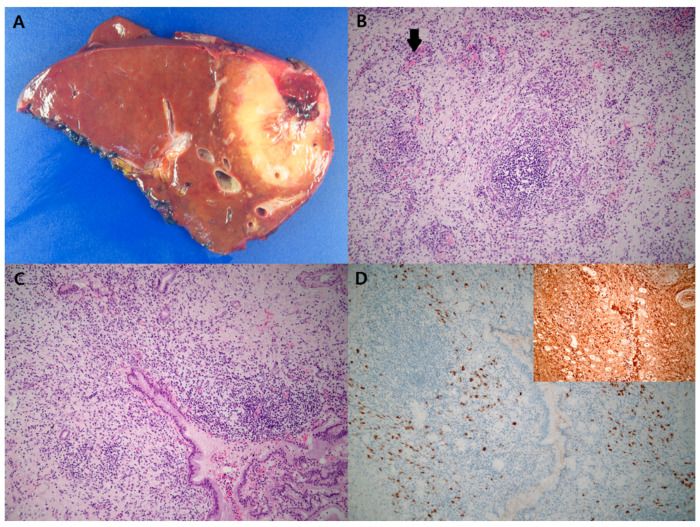
An ill-defined, pale tan to yellow, slightly firm and solid mass is seen, measuring 6.5 × 4.6 × 4.2 cm in maximum dimensions in the right lobe of the liver (**A**). Diffuse lymphoplasmacytic infiltrate with lymphoid aggregates in the fibrotic background. Obliterative phlebitis is also seen (arrow) (**B**, Hematoxylin & eosin stain, ×100). The bile duct is surrounded by a dense lymphoplasmacytic infiltrate with a feature of luminal narrowing. Storiform fibrosis is found in a background. (**C**, Hematoxylin & eosin stain, ×100). An immunohistochemical stain for IgG4 shows a diffuse increase in IgG4+ plasma cells (**D**, IgG4 immunohistochemistry, ×100). The IgG4: IgG ratio is greater than 40% (**Box D**, IgG immunohistochemistry, ×200).

## Data Availability

Not applicable.

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
