# Peer review of "A Case of Hepatic Immunoglobulin G4-Related Disease Presenting as an Inflammatory Pseudotumor and Sclerosing Cholangitis"

_diagnostics, 2022, doi:10.3390/diagnostics12061497_

Round 1

Reviewer 1 Report

In the manuscript titled "A case of hepatic ImmunoglobulinG4-related disease presenting as an inflammatory pseudotumor and sclerosing cholangitis" Se Young Jang and colleagues, they have reported that 

 An inflammatory pseudotumor is a benign disease characterized by tumor-like lesions consisting of inflammatory cells including plasma cells, and fibrous tissue. Recently, some inflammatory pseudotumor cases proved to be a form of ImmunoglobulinG4-related disease (IgG4-RD). This novel clinical entity, recognized as a fibroinflammatory condition, is characterized by lymphoplasmacytic infiltration with a predominance of IgG4-positive plasma cells, storiform fibrosis, and often elevated serum IgG4 concentrations. We report a case of IgG4-RD in the form of an inflammatory pseudotumor in the liver with combined sclerosing cholangitis. We recommend that for diagnosing IgG4-RD accurately, it is important to obtain adequate tissue samples and follow up on the lesion in clinical practice.  I have a few comments regarding the present images.

-A better description of IgG4-RD is necessary and how this new entity has an impact on host health.

-Maybe other diagnostic tests or results should be given to clarify the specific IgG4-RD.

-Some concluding remarks are required. 

Author Response

Reviewer1

In the manuscript titled "A case of hepatic ImmunoglobulinG4-related disease presenting as an inflammatory pseudotumor and sclerosing cholangitis" Se Young Jang and colleagues, they have reported that  An inflammatory pseudotumor is a benign disease characterized by tumor-like lesions consisting of inflammatory cells including plasma cells, and fibrous tissue. Recently, some inflammatory pseudotumor cases proved to be a form of ImmunoglobulinG4-related disease (IgG4-RD). This novel clinical entity, recognized as a fibroinflammatory condition, is characterized by lymphoplasmacytic infiltration with a predominance of IgG4-positive plasma cells, storiform fibrosis, and often elevated serum IgG4 concentrations. We report a case of IgG4-RD in the form of an inflammatory pseudotumor in the liver with combined sclerosing cholangitis. We recommend that for diagnosing IgG4-RD accurately, it is important to obtain adequate tissue samples and follow up on the lesion in clinical practice.  I have a few comments regarding the present images.

  1. A better description of IgG4-RD is necessary and how this new entity has an impact on host health.

-> Thank you for your valuable comment to improve the article. We added the description of IgG4-RD in Introduction section, lines 42-46.

-> IgG4-RD is a novel clinical entity recognized as a fibroinflammatory condition, characterized by lymphoplasmacytic infiltration with a predominance of IgG4-positive plasma cells, storiform fibrosis, and often elevated serum IgG4 concentrations.[5] IgG4-RD can affect multiple organs, [6,7] such as the pancreas, retroperitoneum, kidneys, and bile ducts. It presents as a multi-organ disease and often confused with malignancy, infection, or other immune-mediated conditions. [8] IgG4- RD can lead to organ dysfunction and organ failure due to late recognition and diagnosis of the disease. [9]

  1. Maybe other diagnostic tests or results should be given to clarify the specific IgG4-RD.

-> Thank you for your critical comment. Unfortunately, we had not suspected the IgG4 related disease until the pathologic findings were observed after liver resection. His serum IgG4 level was checked after resection, and that was 749.1 mg/L (reference: 30-2010). It could be in the middle of falling or the initial serum IgG4 level was in a normal range. It is known that elevated serum concentrations of IgG4 are found in 60 to 70 percent of patients with IgG4-RD (Ann Rheum Dis 2015;74:14–8; Medicine (Baltimore) 2016;95:e3785). In addition, recently, the presence of an elevated serum IgG4 level is no longer considered essential to the diagnosis of IgG4-RD (Arthritis Rheumatol. 2020 Jan;72(1):7-19).

We added the level of the serum IgG4 in lines 118-119 according to the reviewer 1 and 2's comments.

-> The serum IgG4 checked after resection was 749.1 mg/L (reference: 30-2010).

In addition, we inserted the explanation of serum IgG4 level according to the reviewer 1 and 2’s comments in Discussion section, lines 160-164.

-> In this case, serum IgG4 was not elevated after resection. It could be in the middle of falling or the initial serum IgG4 level was in a normal range. It is recently recognized that serum IgG4 levels are not elevated in a substantial percentage of patients diagnosed as IgG4-RD and serum IgG4 elevation is no longer considered essential to the diagnosis of IgG4-RD. [13]

  1. Some concluding remarks are required.

-> Thank you for your comment. We added some remarks as a conclusion, lines 180-181.

We followed the case of IgG4-RD manifesting as a hepatic inflammatory pseudo-tumor diagnosed and resolved by hepatic resection with histopathological examination. In conclusion, it is important to doubt IgG4-RD for mass lesion early and to diagnose IgG4-RD accurately by obtaining adequate tissue samples and following-up the lesion.

Reviewer 2 Report

This is a nice case report describing a case of IgG4-RD in the form of an inflammatory pseudotumor in the liver with combined sclerosing cholangitis. The authors properly discuss the importance to obtain adequate tissue samples and follow-up the lesion in clinical practice.

The manuscript is of interest and of clinical relevance. However, some points deserve further data and should be addressed.

-At the clinical presentation, acute cholangitis was diagnosed. Please add laboratory features at the presentation.

-Please add the value of serum IgG4 at the diagnosis.

-To further improve the manuscript I would suggest recalling the potential multi-organ (pancreas, biliary duct, kidneys, lungs, thyroid, and salivary glands.) involvement of IgG4-related disease as recently well described (Multi-Organ Involvement of Immunoglobulin G4-Related Disease. Gastroenterol. Insights 202112, 350-357).

Author Response

Reviewer2

This is a nice case report describing a case of IgG4-RD in the form of an inflammatory pseudotumor in the liver with combined sclerosing cholangitis. The authors properly discuss the importance to obtain adequate tissue samples and follow-up the lesion in clinical practice.

The manuscript is of interest and of clinical relevance. However, some points deserve further data and should be addressed.

  1. At the clinical presentation, acute cholangitis was diagnosed. Please add laboratory features at the presentation.

-> Thank you for your comment. We added his laboratory findings in the Case Description section, lines 54-59.

His laborary findings showed elevated white blood cell counts [11.2 (reference: 4.8-10.8 ) x 103/µL], elevated aspartate transaminase [128 U/L (reference: ≤40)], elevated alanine transaminase [96 U/L (reference: ≤41)], elevated alkaline phosphatase [180 U/L (reference: 40-129)], elevated gamma-glutamyltransferase [132 U/L (reference: 10-71)], and slightly elevated total bilirubin [1.8 mg/dL (reference: <1.2)].

  1. Please add the value of serum IgG4 at the diagnosis.

-> Thank you for your critical comment. Unfortunately, we had not suspected the IgG4 related disease until the pathologic findings were observed after liver resection. His serum IgG4 level was checked after resection, and that was 749.1 mg/L (reference: 30-2010). It could be in the middle of falling or the initial serum IgG4 level was in a normal range. It is known that elevated serum concentrations of IgG4 are found in 60 to 70 percent of patients with IgG4-RD (Ann Rheum Dis 2015;74:14–8; Medicine (Baltimore) 2016;95:e3785). In addition, recently, the presence of an elevated serum IgG4 level is no longer considered essential to the diagnosis of IgG4-RD (Arthritis Rheumatol. 2020 Jan;72(1):7-19).

We added the level of the serum IgG4 in lines 118-119 according to the reviewer 1 and 2's comments.

-> The serum IgG4 checked after resection was 749.1 mg/L (reference: 30-2010).

In addition, we inserted the explanation of serum IgG4 level according to the reviewer 1 and 2’s comments in Discussion section, lines 160-164.

-> In this case, serum IgG4 was not elevated after resection. It could be in the middle of falling or the initial serum IgG4 level was in a normal range. It is recently recognized that serum IgG4 levels are not elevated in a substantial percentage of patients diagnosed as IgG4-RD and serum IgG4 elevation is no longer considered essential to the diagnosis of IgG4-RD. [13]

  1. To further improve the manuscript I would suggest recalling the potential multi-organ (pancreas, biliary duct, kidneys, lungs, thyroid, and salivary glands.) involvement of IgG4-related disease as recently well described (Multi-Organ Involvement of Immunoglobulin G4-Related Disease. Gastroenterol. Insights 2021, 12, 350-357).

-> Thank you for your comment. We inserted the mentioned article as a reference in the introduction section, lines 42-44.

IgG4-RD can affect many organs in the body, [6,7] such as the pancreas, retroperitoneum, kidneys, and bile ducts.

  1. Nour, E.; Hammami, A.; Missaoui, N.; Bdioui, A.; Dahmani, W.; Ben Ameur, W.; Braham, A.; Ajmi, S.; Ben Slama, A.; Ksiaa, M., et al. Multi-Organ Involvement of Immunoglobulin G4-Related Disease. Gastroenterol Insigh 2021, 12, 350-357, doi:10.3390/gastroent12030033.

Round 2

Reviewer 1 Report

Thank you to the authors for taking into account my previous comments, now the manuscript reads well, no further comments are required

Reviewer 2 Report

The Authors satisfactorily addressed the raised points